# Ball Screw Fault Diagnosis Based on Wavelet Convolution Transfer Learning

**DOI:** 10.3390/s22166270

**Published:** 2022-08-20

**Authors:** Yifan Xie, Chang Liu, Liji Huang, Hongchun Duan

**Affiliations:** 1Key Laboratory of Advanced Equipment Intelligent Manufacturing Technology of Yunnan Province, Kunming University of Science & Technology, Kunming 650500, China; 2Faculty of Mechanical & Electrical Engineering, Kunming University of Science & Technology, Kunming 650500, China

**Keywords:** transfer learning, convolutional neural network, adaptive batch normalization algorithm, fault diagnosis, ball screw

## Abstract

The ball screw is the core component of the CNC machine tool feed system, and its health plays an important role in the feed system and even in the entire CNC machine tool. This paper studies the fault diagnosis and health assessment of ball screws. Aiming at the problem that the ball screw signal is weak and susceptible to interference, using a wavelet convolution structure to improve the network can improve the mining ability of signal time domain and frequency domain features; aiming at the challenge of ball screw sensor installation position limitation, a transfer learning method is proposed, which adopts the domain adaptation method as jointly distributed adaptation (JDA), and realizes the transfer diagnosis across measurement positions by extracting the diagnosis knowledge of different positions of the ball screw. In this paper, the adaptive batch normalization algorithm (AdaBN) is introduced to enhance the proposed model so as to improve the accuracy of migration diagnosis. Experiments were carried out using a self-made lead screw fatigue test bench. Through experimental verification, the method proposed in this paper can extract effective fault diagnosis knowledge. By collecting data under different working conditions at the bearing seat of the ball screw, the fault diagnosis knowledge is extracted and used to identify and diagnose the position fault of the nut seat. In this paper, some background noise is added to the collected data to test the robustness of the proposed network model.

## 1. Introduction

The modern manufacturing industry is developing in the direction of intelligence, automation and complexity, which puts forward higher requirements for the machining accuracy and reliability of manufacturing equipment. CNC machine tools are the most commonly used and typical mechatronics equipment in modern manufacturing. The technical level of CNC machine tool manufacturing is one of the important indicators to measure the level of a country’s economic development, and the total ownership of CNC machine tools is also one of the important indicators to measure the overall level of industrial manufacturing. As one of the important transmission components of CNC machine tools, a ball screw pair directly affects the quality of processed products. This puts forward higher requirements for the fault diagnosis and health assessment of the ball screw pair. At present, the research on the fault diagnosis of the ball screw is basically to process and analyze the data collected at the nut seat of the screw. Korean scholar, Won et al. [1] used the collected vibration signals to diagnose the fault of the ball screw, and developed a fault diagnosis system that can detect the fault severity and fault location of the ball screw; Shan et al. [2] The data collected by sensors installed in different positions were weighted and combined with a convolutional neural network to realize fault diagnosis of ball screws; Yang et al. [3] The current signal of the drive motor of the ball screw is collected, and then extracted, followed by filtering the characteristics of the collected current signal by means of short-time Fourier transform (STFT) and wavelet decomposition (WPD). The feature utilizes logistic regression and K-nearest neighbors to achieve fault diagnosis for ball screws. Although the vibration signal collected at the position of the nut seat in the ball screw pair is better than the position of the support bearing seat, it is not convenient to install the sensor at the nut seat of the ball screw in the actual industrial field. This leads to the fact that if the neural network model obtained by training the data collected at the nut seat is directly used, the diagnostic knowledge obtained through machine learning cannot be directly applied to the fault diagnosis of industrial field ball screws. At the same time, the working conditions of the ball screw in the industrial field are complex and changeable, and the interference is large. Therefore, the fault diagnosis of the ball screw in the industrial field is more difficult.

Facing the problems mentioned above, the theoretical method of transfer learning provides new ideas and approaches. Transfer learning is a new machine learning method that uses pre-existing knowledge to solve different but similar problems. This technical method can solve the learning problems of scarcely available training samples. Transfer learning has been successfully applied in object recognition [4], speech processing [5], text recognition [6] and other fields. With the development of transfer learning theory, more and more people are beginning to use the theoretical knowledge of transfer learning to solve practical problems. In the field of fault diagnosis, there are more and more studies on the fault diagnosis of components using the theoretical knowledge of transfer learning. The research on faulty parts diagnosis based on transfer learning is mainly divided into two parts. One part is the knowledge transfer of fault diagnosis between different working conditions; the other part is the knowledge transfer of fault diagnosis between different locations. The transfer of fault diagnosis knowledge between different working conditions is to use the fault diagnosis knowledge of parts in the original working conditions in the fault diagnosis of the parts under different rotational speeds and load conditions. Cao et al. [7] used CNN to realize the fault diagnosis of gears using a small sample training set; Azamfar et al. [8] The convolutional neural network is used to extract features, and the maximum mean difference measure is used to optimize the data distribution of different working conditions during the training process, and finally, the fault diagnosis of the ball screw is realized. The transfer of fault diagnosis knowledge between different locations is to use the fault diagnosis knowledge of a component in one location in the fault diagnosis of the component in other locations. The main goal of this type of research is to reduce the difference in feature distribution between the source domain data and target domain data. Tong et al. [9] proposed a domain adaptation method with transferable features, using the maximum mean difference (MMD) to reduce the difference between the source domain and the target domain, and then realize the bearing fault diagnosis; Zhang et al. [10] proposed an unsupervised domain adaptation method, which uses subspace alignment to transfer the fault diagnosis knowledge of components in one location to the fault diagnosis of components in other locations. Liao et al. [11] proposed a fault diagnosis method for rotating machinery that uses wavelet decomposition and a combination of energy pooling and convolutional neural networks to obtain the energy features of the signal. Marginal and conditional distribution differences between source and target domain data are reduced by maximum mean difference (MMD) during training so as to complete the diagnosis across operating conditions and locations, and verify that the robustness of the proposed model is significantly higher than that of traditional convolutional neural networks. When the source domain and target domain feature distributions are quite different, Li et al. [12] proposed a domain adaptation method of adaptive batch normalization (AdaBN) based on the batch normalization layer (BN layer), and used the BN layer to endow the neural network with good domain adaptability. Zhang et al. [13] proposed a WDCNN model based on the adaptive batch normalization algorithm, and used AdaBN to improve the domain adaptability of the model. Experiments show that AdaBN has a good adaptive ability for the transfer of fault diagnosis knowledge between different working conditions of the same part. Li et al. [14] added the AdaBN algorithm to the DNN to improve the domain adaptability of the DNN and achieve the effect of deep adaptation on the domain adaptation task.

In this paper, the problem of fault diagnosis and health assessment of a ball screw is studied. Aiming at the problem that the ball screw signal is weak and susceptible to interference, the wavelet convolution structure is used to improve the network to realize the deep mining of the time-frequency domain characteristics of signals; aiming at the challenge of the limited installation position of the ball screw sensor, a transfer learning method based on jointly distributed adaptation (JDA) is proposed to realize the diagnosis knowledge extraction of different positions of the lead screw and the transfer diagnosis across the measurement positions. Aiming at the complex and changeable working conditions of industrial CNC machine tools, the proposed model is improved by using the adaptive batch normalization algorithm (AdaBN) to improve the accuracy of a migration diagnosis. In this paper, the proposed method is experimentally validated using a self-made lead screw fatigue test rig. The experimental results show that the method can extract diagnostic knowledge from the data collected from the support bearing seats at both ends of the lead screw under different working conditions and realize the identification and diagnosis of the position fault of the nut seat through knowledge transfer. The model proposed in this paper has good robustness under noise interference.

## 2. Theoretical Background

### 2.1. Wavelet Convolution Energy Pooling

The constructed convolutional neural network based on the domain adaptation algorithm AdaBN model (AdaCNN) enhances the associated fault features by constructing wavelet convolution, and the convolution kernel (filter kernel) performs convolution operation on the input vibration signal to extract features. In signal processing, time domain convolution corresponds to frequency domain multiplication. The convolution of the vibration signal with the convolution kernel actually extracts the frequency domain information. The information extracted by the convolution layer of the one-dimensional CNN through the convolution operation is the different frequency domain information of the vibration signal. The high-level convolution layer extracts high-level features, and the convolution essentially filters the vibration signal. The output calculation of neurons in the convolutional layer is shown in Equation (1):(1)$$g(i)=\sum_{x=1}^{m}\sum_{y=1}^{n}\sum_{z=1}^{p}a_{x,y,z}\times w_{x,y,z}^{i}+b^{i}$$

In Formula (1), i represents the ith convolution kernel (layer i), g(i) represents the feature map obtained by the ith convolution kernel, $a_{x,y,z}$ represents the value of the node (x,y,z) in the convolution kernel and $b^{i}$ represents the bias of the convolution kernel.

The wavelet analysis method performs a convolution operation on the signal f(t), and the convolution kernel passes through the different scale basis functions ψ(f,t) of the wavelet function selected for determination. It obtains ideal analysis results by constructing appropriate basis functions. In digital signal processing, the convolution can be expressed as shown in Equation (2). Among them, x[n],h[n] is the discrete sequence.
(2)$$y[n]=x[n]*h[n]=\sum_{k=-\infty }^{\infty }x[k]\cdot h[n-k]$$

On the basis of classic CNN, this paper designs wavelet groups based on original wavelets, initializes the first layer of convolution kernels with wavelet kernels of different scale bases and uses the ability of wavelets to mine time-frequency domain features of signals to extract fault features at relevant locations. The representation method of the wavelet convolution layer is shown in Equation (3):(3)g(i)=x∗ψ(f,t)

This paper constructs a convolution kernel with different scale bases based on the Daubechies wavelet. The ability of a wavelet to extract time-frequency domain features of vibration signals is used to obtain effective correlated fault features, and gradient descent backpropagation is allowed to update them during training.

In the fault diagnosis of rotating machinery, the root mean square value is often used to represent the energy of the vibration signal because it has better stability. The root mean square value is an important indicator for judging whether the equipment is running normally and for diagnosing the wear of equipment parts and other faults. For the one-dimensional vibration signal x, the RMS energy calculation formula is shown in Equation (4):(4)$$X_{rms}=\sqrt{\frac{\sum_{i=1}^{N}x_{i}^{2}}{N}}$$

This paper proposes an energy pooling method, and the calculation method is shown in Equation (5):(5)$$f(X)=\sqrt[{p}]{\frac{\sum_{x\in X}x^{p}}{N}}$$

The max pooling in the pooling operation is equivalent to when p equals infinity, and the average pooling in the pooling operation is equivalent to when p = 1. This paper takes p = 2. The comparison of the results of the three pooling methods of max pooling, average pooling and energy pooling is shown in Figure 1. The extracted energy pooling method is more reasonable.

In this paper, the convolutional neural network is used to replace the convolutional layer with a wavelet convolutional layer and the pooling layer with an energy pooling layer, thereby improving the model’s ability to mine vibration signal features so as to solve the problem of weak failure of the CNC machine tool feed system at actual industrial sites.

### 2.2. AdaBN Algorithm

Google researchers, Szegedy and Ioffe [15] proposed a batch normalization algorithm that calculates the mean and standard deviation of a small batch of samples of the input signal. This ensures that under different batches of samples, the input distribution of each layer is basically unchanged, which can increase the training speed and effectively prevent the phenomenon of gradient disappearance or gradient explosion. When the distribution of features in the training set (source domain) and the test set (target domain) is quite different, the accuracy of the test set and the accuracy of the training set will differ greatly. Li et al. [12] proposed the domain adaptation method of adaptive batch normalization (AdaBN), which improves the domain adaptability of the network by adjusting the information of the BN layer. The AdaBN algorithm is adapted to situations where the features of the source domains and target domains are similar in distribution.

In order to deal with the complex and changeable machining conditions of CNC machine tools in the current industrial field, the AdaBN algorithm is added in this paper to increase the domain self-adaptation ability of the network. The AdaBN algorithm is mainly divided into three parts: first is to use the training set (source domain) samples to input the neural network to calculate the statistical information of all BN layers (mean, standard deviation, scaling parameters and translation parameters); second is to input the test set (target domain) into the neural network to compute the statistical information (mean and standard deviation) of all BN layers; finally, the statistics of each BN layer of the test set are to be replaced by the statistics of the BN layers in the source domain. Therefore, the features extracted from the test set data and the training set data are basically the same in distribution, which improves the domain adaptability of the network model. The specific algorithm flow of AdaBN is shown in Algorithm 1. The preparation of AdaBN is divided into the following parts: The vibration signal is processed in segments and divided into a training set (source domain) and a test set (target domain); the training set is input into neural network training, and the scaling parameters and translation parameters are obtained.
**Algorithm 1:** AdaBN AlgorithmInput: all *m* samples *x* in test set *t*, scaling parameters and translation parameters of training setFor the *i*th neuron of the neural network:Calculate the mean: μit=E(xit)Calculate variance: δit=Var(xit)Calculate the output of the BN layer: yi(m)=γi(xi(m)−μit)δit+βiend for

### 2.3. Joint Distribution Adaptation (JDA) Feature Domain Adaptation

The joint distribution adaptation (JDA) method is used to calculate and reduce the joint probability distribution distance between the source domain data and the target domain data. This method calculates the edge probability distribution and conditional probability distribution between the source domain and the target domain. The calculation formula of the conditional probability distribution MMD distance is shown in Equation (6):(6)$$\mathrm{MMD}(Q_{s}^{c},Q_{t}^{c})={\left\|\frac{1}{n_{s}^{c}}\sum_{x_{i}^{s}\in D_{s}^{c}}^{n_{s}}\phi (x_{i}^{s})-\frac{1}{n_{t}^{c}}\sum_{x_{j}^{t}\in D_{t}^{c}}^{n_{t}}\phi (x_{j}^{t})\right\|}_{H}^{2}$$

The JDA consists of two parts, the joint marginal distribution distance and the conditional distribution distance, respectively, and the calculation formula of its regular term is shown in Equation (7):(7)$$D(J_{s},J_{t}{)=\mathrm{MMD}}_{H}^{2}(P_{s},P_{t})+\sum_{c=1}^{C}{\mathrm{MMD}}_{H}^{2}(Q_{s}^{C},Q_{t}^{C})$$

The minimum value of the above formula can ensure that both the marginal probability distribution and the conditional probability distribution have sufficient statistical information. At this time, the optimization objective of the model is expressed, as shown in Equation (8):(8)$$\underset{\theta }{argmin}\frac{1}{n_{a}}\sum_{i=1}^{n_{a}}J(\theta (x_{i}^{a}),y_{i}^{a})+\lambda D(J_{s},J_{t})$$

In Equation (8), θ represents all the weights and bias parameters of the network, which is the target of model learning; λ is the penalty coefficient; J() defines the loss function. In this paper, the loss function uses the cross-entropy loss function.

In this paper, JDA is selected as the domain adaptation method for the transfer of fault diagnosis knowledge between different locations. The specific implementation method performs forward propagation during the training transfer, calculates the cross-entropy loss function and the regular term of JDA, and then uses the backpropagation algorithm and Adam optimization algorithm to adjust model parameters. The advantage of doing this is that, on the one hand, by optimizing the loss function, the model parameters are adjusted to achieve the correct identification of the source domain data. On the other hand, by optimizing the JDA regularization term, the neural network can further reduce the difference in feature distribution between the source and target domains and learn the representation of domain-invariant features from it. Thus, the diagnostic knowledge learned in the source domain is applied to the target domain data.

## 3. Fault Diagnosis Method Based on AdaCNN Model

A transfer learning model based on wavelet convolution (AdaCNN) is proposed based on a convolutional neural network. The model uses wavelet decomposition to extract the time-frequency domain features of vibration signals and obtains the energy features of signal channels through energy pooling. The difference between the domains is reduced by the domain adaptation method in the model. The fault diagnosis of the ball screw under different operating conditions and from different positions is finally realized. The model mainly includes two parts: the transfer of fault diagnosis knowledge between different working conditions and the transfer of fault diagnosis knowledge between different locations.

### 3.1. A fault Diagnosis Knowledge Transfer Method between Different Working Conditions Based on AdaBN Algorithm

Due to the complex and changeable working conditions of CNC machine tools in the current industrial field, this paper proposes a ball screw fault diagnosis method based on transfer learning. This method adopts the adaptive batch normalization algorithm (AdaBN) to increase the domain adaptive ability of the model, and realizes the transfer of fault diagnosis knowledge between different working conditions at the ball nut seat. In the actual working conditions, the rotational speed of the lead screw is relatively fixed, and the loads generated by different workpieces are basically different. Therefore, different working conditions in this paper refer to different loads.

Figure 2 is the flowchart of the method proposed in this section. The main steps of the method are as follows:Collect the vibration data under different loads at the ball screw nut seat, divide the sample source and target domains with the load as a variable, and divide the source domain samples into training set samples and test set samples according to a certain proportion.Build a convolutional neural network. In Figure 2, we show the structure of a convolutional neural network. The first convolutional layer is replaced with a wavelet convolutional layer, and the pooling layer is replaced with an energy pooling layer. The improved convolutional layer and pooling layers can improve the model’s ability to mine signal features. A batch normalization layer (BN) is added after the convolutional layer, and the training process adopts mini-batch learning. Batch normalization (BN) is to transform the input data distribution into a normal distribution with a mean equal to 0 and a variance equal to 1. The Adam optimization algorithm is used when the parameters are updated, which is beneficial for accelerating the training speed. Train and save the model parameters as Model 1, with the target domain sample training set.Fix the BN layer parameters of Model 1, use the source domain for training, only perform forward propagation during the training process, and save the model parameters as Model 2 after training. At this time, the role of the BN layer makes the source domain and the target domain similar in feature distribution, thereby improving the generalization ability of the model.Input the test set of target domain samples into Model 2 to verify the correctness of the network.

### 3.2. Fault Diagnosis Knowledge Transfer Method between Different Locations Based on Joint Distribution Adaptation (JDA)

There is an important problem in the current industrial field. It is inconvenient to install sensors at the nut seat of the ball screw of the CNC machine tool. Aiming at this problem, this paper proposes a method for the transfer of fault diagnosis knowledge between different locations. During the transfer process, joint distribution adaptation (JDA) is used to calculate and reduce the difference between the marginal distribution and conditional distribution of the source domain and the target domain to form a transfer model. The method realizes the transfer of fault diagnosis knowledge at the nut seat to the bearing seat. Figure 3 shows the flowchart of the method proposed in this section, and the main steps of the method are as follows:Under the same working conditions, the data collected at the nut seat of the ball screw (the target domain sample in the fault diagnosis knowledge transfer between different working conditions) are used as the source domain data.Build the convolutional neural network model. The structure of the convolutional neural network is shown in Figure 3. The first convolutional layer is replaced with a wavelet convolutional layer, and the pooling layer is replaced with an energy pooling layer.On the basis of Model 2, the JDA algorithm matching feature is added, and it is added to the optimization goal of the network. JDA calculates the domain difference through forward propagation, and backpropagation reduces the domain difference. In the training process, the model parameters are fine-tuned with the target domain dataset, and the BN layer parameters are fixed to make the feature distribution of the source domain and the target domain basically the same.

### 3.3. AdaCNN Model

The overall structure of the AdaCNN model consists of five one-dimensional convolutional layers, two pooling layers, and three fully connected layers. AdaCNN adds a normalization layer after the convolution layer. On the one hand, it can effectively solve the problem of vanishing gradient or gradient explosion during training; on the other hand, when migrating fault diagnosis knowledge, it can make the source domain and target domain similar in feature distribution and improve the generalization ability of the model. AdaCNN adds a normalization layer after the convolution layer. On the one hand, it can solve the problem of gradient disappearance or gradient explosion during training; on the other hand, when the fault diagnosis knowledge is transferred, the feature distribution of the source domain and the target domain can be similar, and the generalization ability of the model can be improved. In this paper, the knowledge learned in the source task is transferred to the new task by fixing the BN layer, and finally, the fault diagnosis of the ball screw is realized.

AdaCNN improves the traditional convolutional neural network through wavelet convolution and energy pooling and enhances the model’s ability to mine features. AdaCNN realizes the fault diagnosis of the ball screw by transferring the fault diagnosis knowledge twice. The first time is the transfer of fault diagnosis knowledge at the ball screw nut under different working conditions. The second time is to transfer the fault diagnosis knowledge from the nut seat to the support bearing seat away from the motor end. This method can solve the problems of weak fault characteristics of the feeding system of the numerical control machine tool, complex and changeable processing conditions, and good vibration signals at the nut seat of the ball screw pair, but it is inconvenient to install the sensor.

Since the feature distributions extracted by the model are relatively consistent between different working conditions and tasks, this paper uses the AdaBN algorithm to improve the model and to improve the generalization ability of the model. During the operation of the ball screw, the nut seat (working platform) reciprocates along the screw, and the distance between the nut seat and the supporting bearing seat has different effects on the bearing seat. Therefore, the fault diagnosis knowledge at the nut seat of the ball screw is transferred to the support bearing seat, which is equivalent to the change of working conditions and positions. During the transfer process, this paper utilizes a fixed BN layer to make the source and the target domains have similar domain distributions. At the same time, the JDA algorithm is added to the fully connected layer and calculates the difference between the source and target domains via forward propagation. The backpropagation reduces the domain difference and fine-tunes the parameters of other layers. The fault diagnosis knowledge learned from the data collected at the nut seat of the ball screw is transferred to the fault diagnosis of the data collected at the bearing seat, and finally, realizes the fault diagnosis of the ball screw. The model in this paper selects the convolutional neural network model, including the convolutional layer, the pooling layer and the fully connected layer. The BN layer is added after the convolutional layer. The first convolutional layer is replaced with a wavelet convolutional layer, and the pooling layer is replaced. For energy pooling, a dropout is added after the second convolutional layer and the fourth convolutional layer to prevent overfitting. The specific parameters of the model are shown in Table 1.

## 4. Ball Screw Experimental Verification

### 4.1. Diagnostic Dataset Description

In this paper, the self-made ball screw test bench is used to collect the vibration signals at the ball screw nut seat (working platform) and the bearing end away from the motor side. The self-made ball screw experiment is shown in Figure 4. On the one hand, the working platform is connected to the nut seat through screws, and on the other hand, it is connected to the guide rail through the slider. The guide rail selects the sliding guide rail. The motor drives the ball screw to rotate, so that the working platform slides along the guide rail. The manufacturing method of wear failure is to use a file to process in the raceway of the screw, and the failure is shown in Figure 4. This experiment uses an acceleration sensor (model: ICP603C01) to collect acceleration vibration signals under various working conditions with a sampling frequency of 20 k, and the acquisition card is NI9234. In this experiment, vibration information was collected at the ball screw nut seat (working platform) and at the bearing seat far away from the motor end. The specific location of the sensor installation during collection is shown in Figure 4. In this experiment, data on normal and fault states were collected. The data collected at the nut seat (working slide) include: the working condition where the load is 0 kg and the speed is 300 r/min (A1), where the load is 15 kg and the speed is 300 r/min (A2), where the load is 0 kg and the speed is 1200 r/min (A3) and where the load is 15 kg and the speed is 1200 r/min (A4). The data collected at the bearing seat include: where the load is 15 kg and the speed is 300 r/min (B1) and where the load is 15 kg and the speed is 1200 r/min (B2). The data information is shown in Table 2. The collected dataset was divided into a training sample set and a test sample set according to the ratio of 80 to 20%. This experiment uses the above data to train and verify the proposed method. In the experiment, the number of training set and test set samples were 1600 and 400, respectively, and the number of data points in each set was 8192.

### 4.2. Transfer of Fault Diagnosis Knowledge between Different Working Conditions

Aiming at the method proposed in this paper, the designed diagnostic task is to realize the load from 0 to 10 kg at the nut seat of the ball screw at the screw speed of 300 r/min, that is, from A1–2, under different working conditions, and transfer troubleshooting knowledge between them. The selection method of the hyperparameters of the experimental model is the most commonly used trial-and-error method. During training and transfer, the batch_size was set to 256, the learning rate was set to 0.01, and the decay rate was set to 0.99.

To verify the effectiveness of the fault diagnosis knowledge transfer method proposed in this paper for the fault diagnosis of the ball screw under different working conditions, several experiments were carried out on the dataset collected at the nut seat of the ball screw. To further illustrate the advantages of the proposed method, the proposed method was compared with the standard CNN method and the WDTL model mentioned in [11]. At the same time, this experiment was also compared with the shallow machine learning method, and the support vector machine (SVM) classifier was selected, and the kernel function of SVM was the radial basis function. RMS, kurtosis, peak, impulse factor, skewness, shape factor, crest factor and margin and the eight eigenvalues were the classification features used by the SVM. Secondly, the robustness of the proposed method was verified by comparing the analysis results of noisy signals.

The traditional transfer learning method was fixed and fine-tuned, that is, the first few layers of the model were fixed during migration, and the parameters of the other layers were fine-tuned. This paper compares the WDTL model [11] with the first four layers fixed using the feature domain adaptation method maximum mean difference (MMD) and the standard CNN model, in which the parameters of the first four layers are fixed, and the feature domain adaptation method is joint distribution adaptation (JDA). When migrating, iterating 50 times to get the average correct rate is required. Additionally, compared with the shallow learning method SVM, Figure 5 shows the results of several methods. It can be seen that the accuracy of the proposed method is obviously better than other methods when transferring under different working conditions.

In order to verify the robustness of the model, this paper compares the three deep learning methods in the case of adding noise, 4 db, the average accuracy rate of 50 iterations, and SVM comparison. Table 3 shows the accuracy of knowledge transfer of several methods under different working conditions and with 4 db of noise added. It can be seen that the diagnosis of the model proposed in this paper is still suitable for high-intensity noise environments, while in the other three models, the average diagnostic accuracy is approximately 98%. This shows that the AdaCNN algorithm can handle signals containing noisy noise, and the model has good robustness against noise.

In order to verify the generalization performance of AdaCNN, this experiment designed the task as A1–A2 datasets and added 4 db noise, and then compared the ability of each convolutional layer of AdaCNN and WDTL [11] to mine data-related features. This experiment used the t-SNE method to visually analyze the features extracted by the model. The visualization results of the second convolution layer are shown in Figure 6, where label 1 represents normal data, and label 0 represents fault data. It can be seen that AdaCNN and WDTL have relatively poor clustering effects, but the classification errors are small, which more intuitively shows the effectiveness of AdaCNN.

### 4.3. Transfer of Fault Diagnosis Knowledge from Different Locations

For the method proposed in this paper, the designed diagnostic task is that the ball screw speed is 300 r/min, the load is 15 kg and knowledge transfer of fault diagnosis from the nut seat of the ball screw to the support bearing seat far away from the motor end, that is, from A2-B1, the transfer of knowledge between different measuring points. The model used in the transfer is the model obtained across the working conditions. The hyperparameters are obtained by the trial-and-error method, batch_size is set to 128, and the learning rate is 0.0001.

In order to verify the effectiveness of the proposed method of fault diagnosis knowledge transfer between different measuring points proposed in this paper, we conducted multiple experiments on datasets collected on ball screws. This experiment also compared different neural network models (standard CNN, WDTL) and shallow learning methods SVM. The classification features used by SVM are the same as those used across working conditions. In the same way, 50 iterations were performed to take the average correct rate. Figure 7 shows the results. As can be seen, the effect of WDTL is very poor, or even lower than that of SVM, and there is little difference between AdaCNN and standard CNN during transmission.

Similarly, this experiment added 4 db noise to the original dataset to verify the robustness of the model. The results are shown in Table 4. It can be seen that although the average accuracy of AdaCNN is 95.32% compared with other models, the highest accuracy of AdaCNN can reach 99.5%. Comparing several other models is enough to illustrate its antinoise performance.

In order to verify the generalization performance of AdaCNN, this experiment designed the task for the A2-B2 dataset and added 4 db of noise. Due to the poor WDTL effect, this experiment compared the ability of AdaCNN and standard CNN to mine data-related features at the second convolution layer. This experiment used the t-SNE method to visually analyze the features extracted by the model. The result is shown in Figure 8. Where label 1 represents normal data, and label 0 represents fault data. It can be seen from the results that AdaCNN has a better and more stable feature distribution effect.

In order to illustrate the necessity of transfer learning, this paper directly classifies the data collected at the bearing end, and uses SVM and standard CNN for training, respectively. The number of iterations is the average accuracy rate within 50 times. This experiment collected the normal and fault data of the bearing end under 300 r/min and 1200 r/min with a 15 kg load, respectively, that is, the B1 and B2 datasets were directly classified and transferred learning (A1-A2-B1, A3-A4-B2) for comparison. The results are shown in Figure 9, and it can be seen that the transfer has a higher accuracy than the direct classification.

## 5. Conclusions, Limitations, and Future Research

In this paper, a ball screw feature enhancement method based on a wavelet convolution transfer learning model is proposed. It solves the problems that the ball screw signal is weak and easily disturbed, the processing conditions are complex and changeable, and the vibration signal at the ball screw pair nut seat is good, but it is not convenient to install the sensor. The proposed method works directly with the data of different working conditions at the nut end as the input of the model, mines the fault features of the associated signals through the wavelet convolution kernel and obtains the energy features of the signal channel by energy pooling using the AdaBN algorithm to improve the adaptability of the model and realize the transfer of diagnostic knowledge between different working conditions. Combined with the JDA algorithm to reduce the difference in domain distribution, enhance the feature transferability in the task-specific layer of the neural network, and complete the transfer of diagnostic knowledge between different locations, it realizes the extraction of diagnostic knowledge from the data collected from the support bearing seats at both ends of the lead screw, and realizes the identification and diagnosis of the position fault of the nut seat through knowledge transfer. Meanwhile, it also has good robustness under strong background noise. In the research process of this paper, the rotational speed fluctuation, lubrication conditions, oil film thickness and other factors of the ball screw during operation were not considered, and the fault type is relatively simple, which will be the main content of our next research work.

## Figures and Tables

**Figure 1 sensors-22-06270-f001:**
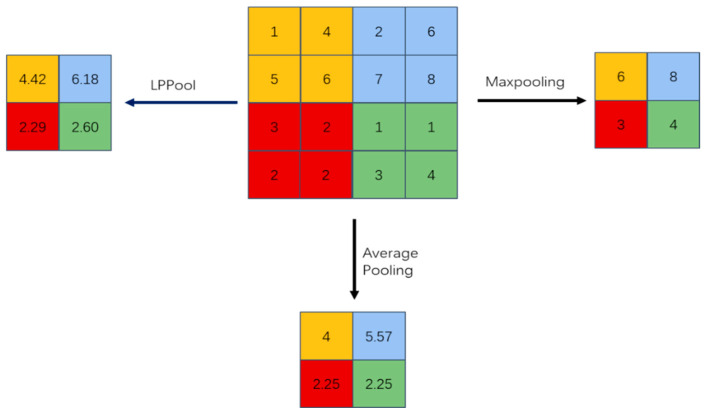
Comparison of the results of three pooling methods: max pooling, average pooling and energy pooling.

**Figure 2 sensors-22-06270-f002:**
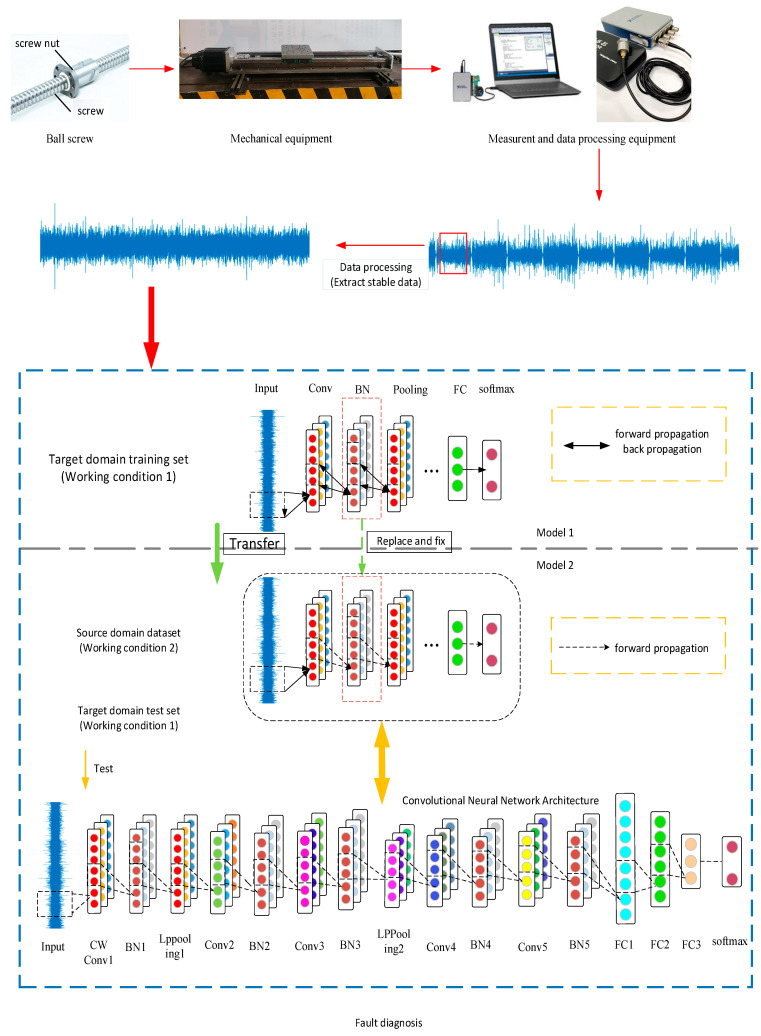
Transfer of fault diagnosis knowledge between different working conditions.

**Figure 3 sensors-22-06270-f003:**
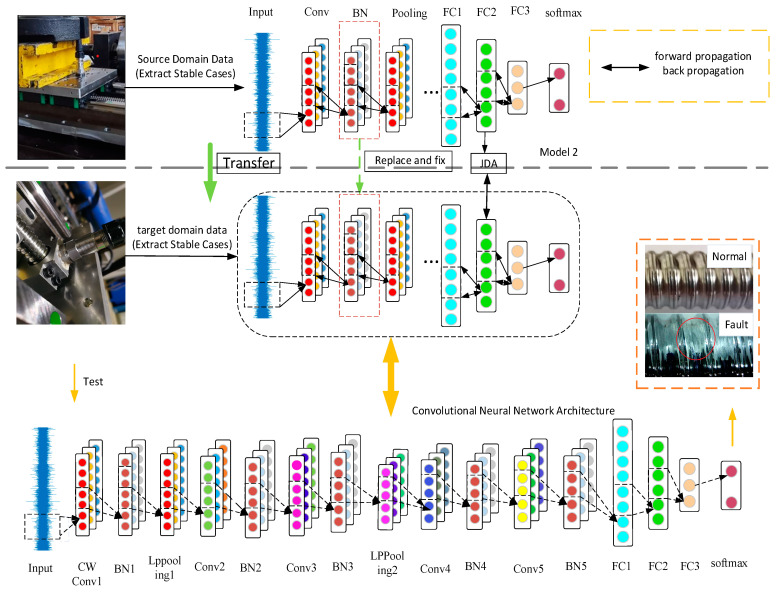
Transfer of fault diagnosis knowledge between different locations.

**Figure 4 sensors-22-06270-f004:**
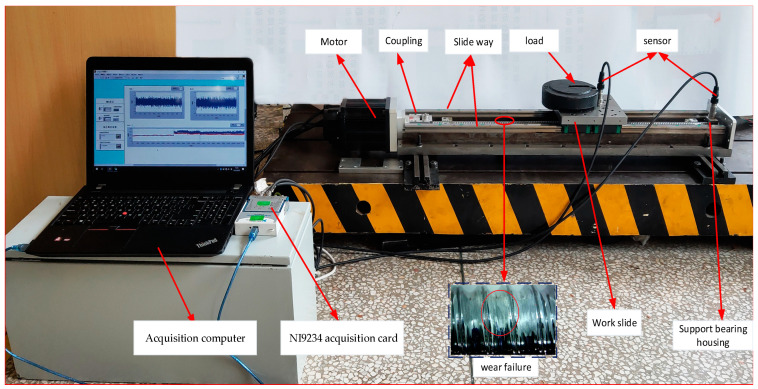
Ball screw test bench.

**Figure 5 sensors-22-06270-f005:**
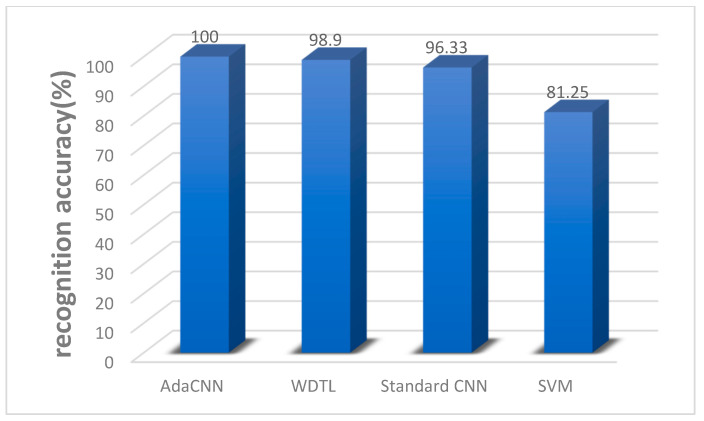
Accuracy of knowledge transfer under different working conditions.

**Figure 6 sensors-22-06270-f006:**
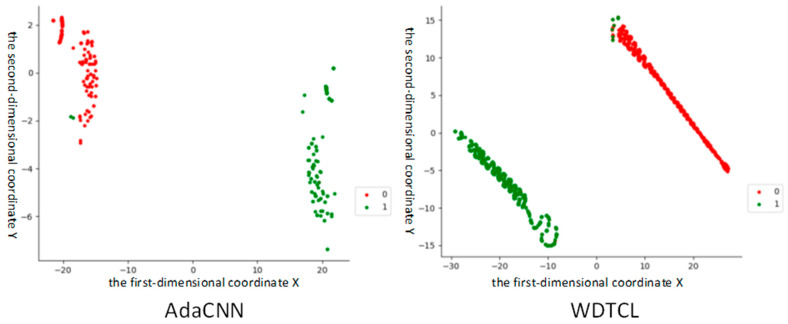
Comparison of visualization effects of the second convolutional layer when fault diagnosis knowledge is transferred between different working conditions.

**Figure 7 sensors-22-06270-f007:**
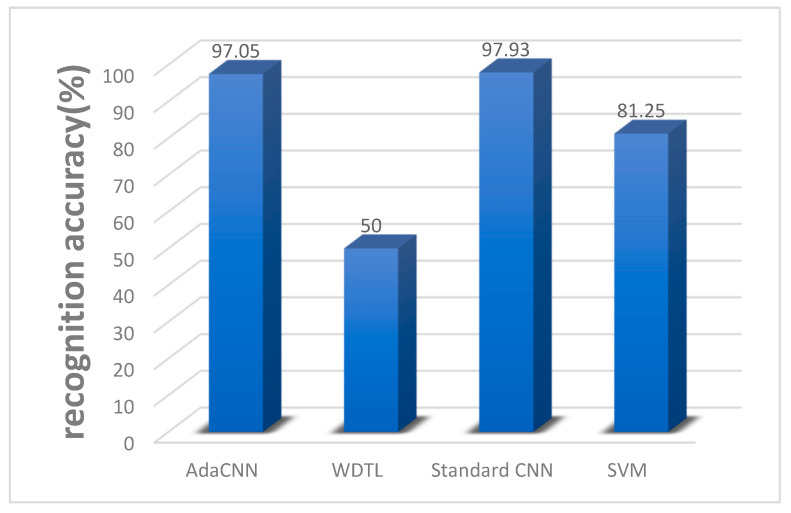
The correct rate of fault diagnosis between different locations.

**Figure 8 sensors-22-06270-f008:**
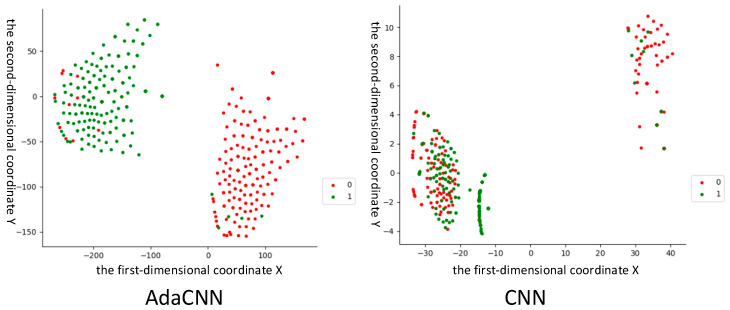
Comparison of visualization effects of the second convolutional layer when fault diagnosis knowledge is transferred between different locations.

**Figure 9 sensors-22-06270-f009:**
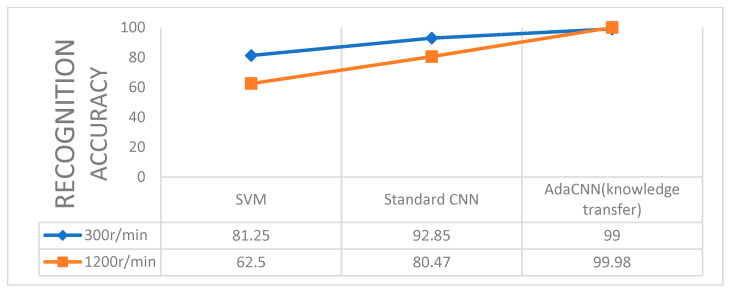
Comparison of transfer learning and direct classification accuracy.

**Table 1 sensors-22-06270-t001:** Model parameter table.

Numbering	Layer	Nuclear Size	Activation Function	Output Size
1	WaveletConv1	1 × 27 × 55	Relu	8192 × 27
2	Pooling1	16 × 1		
3	Conv2	27 × 27 × 55	Relu	512 × 27
4	Dropout			
5	Conv3	27 × 27 × 55	Relu	512 × 27
6	Pooling2	16 × 27		
7	Conv4	27 × 27 × 55	Relu	32 × 27
8	Dropout			
9	Conv5	27 × 27 × 55	Relu	32 × 27
10	Flatten			864 × 1
11	Full1	864 × 216	Relu	864 × 1
12	Full2	216 × 64	Relu	64 × 1
13	Full3	64 × 2	Softmax	2 × 1

**Table 2 sensors-22-06270-t002:** Collected data information.

load	rotating speed	300 r/min	1200 r/min
	position	train/test set	train/test set
0 kg	nut seat (work platform)	A1	A3
15 kg	nut seat (work platform)	A2	A4
	bearing housing	B1	B2

**Table 3 sensors-22-06270-t003:** The accuracy of knowledge transfer, in the case of adding noise 4 db, under different working conditions.

	Accuracy
AdaCNN	100%
WDTL	98.9%
Standard CNN	98.5%
SVM	75%

**Table 4 sensors-22-06270-t004:** The correct rate of fault diagnosis in different positions under the condition of adding 4 db of noise.

	Accuracy
AdaCNN	95.32%
WDTL	50%
Standard CNN	93.48%
SVM	75%

## Data Availability

The data presented in this study are available on request from the corresponding author.

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
