# Peer review of "Ball Screw Fault Diagnosis Based on Wavelet Convolution Transfer Learning"

_sensors, 2022, doi:10.3390/s22166270_

Round 1

Reviewer 1 Report

Aiming at the problem that the ball screw signal is weak and susceptible to interference, the wavelet convolution structure is used to improve the network. In view of the challenge of the installation position limitation of the ball screw sensor, the domain adaptation method of Joint Distributed Adaptation (JDA) is adopted. , enabling migration diagnostics across measurement locations by extracting diagnostics. Through the experimental data under different working conditions

Its effectiveness and superiority are verified. The main features and achievements of this paper are as follows:

1. Aiming at the challenge of limited installation positions of ball screw sensors, a transfer learning method based on Joint Distributed Adaptive (JDA) is proposed to realize the extraction of diagnostic knowledge at different positions of the screw and the transfer diagnosis across measurement positions.

2. In view of the complex and changeable working conditions of industrial CNC machine tools, the adaptive batch normalization algorithm (AdaBN) is used to improve the proposed model to improve the accuracy of migration diagnosis.

In general, I think this paper has a certain contribution to the field of fault diagnosis, but the paper needs to be revised. One main suggestion is as follows: It is recommended to delete the part about CWRU at the end of the paper, which is not related to the theme of the paper. Some other suggestions are as follows:

1. "This paper takes 165 p=2" in p4. Proper explanation is recommended for the value of p.

2. min in formula 8 of line 214 of P5 should be argmin.

3. The cross loss function in line 217 of P5 should be Cross Entropy Loss Function

4. The preparations in the pseudo-algorithm table on page 5 should be stated outside the table

5.P6 Line 244 The sentence breaks semantically unreasonable.

6. P7 and Figure 1 and Figure 2 on page 8 can be adjusted, try to describe it with one picture, the repetition of the two pictures is a bit high

7. In P10, the data in the training set and test set is less and it is recommended to increase.

8. The layout of page 10 should be rearranged, and the fourth page should be placed in Table 3

9. Table 3 on pages 10 and 11, and table 7 on pages 15 and 16 can be completely on one page, so that the tables are not separated; in addition, the figure and figure name of figure 4 on page 11 are separated on two pages

10.P12 Line 399 should be AdaCNN according to Figure 4 and the preceding text.

11.P12 410 lines AdaCNN in the text and AdaBN in Figure 5.

12.P12 Line 419 The previous gradient descent method is batch gradient descent, and here is mini-batch gradient descent. It is recommended to give a brief reason for the choice.

13.P13 Line 433 Although deleted, there is no turning point in this sentence.

14. In p14, "This experiment collected the normal and fault data of the bearing end under 300r/min and 1200r/min" only selects 300r/min and 1200r/min, is it also effective at higher speeds.

15.P16 Line 498 The limitations in the conclusion are not mentioned, and the description of future development is too abstract.

16. Page 17, no punctuation at the end of reference 15

Author Response

Dear Reviewer:

Thank you very much for giving us an opportunity to revise our manuscript. We greatly appreciate your constructive comments and suggestions on our manuscript sensors-1850320. The title of the manuscript is ’Ball screw fault diagnosis based on wavelet convolution transfer learning’. Those comments are very helpful for revising and improving our paper, as well as the important guiding significance to other research. We have studied the comments carefully and made corrections which we hope meet with approval. Major corrections are made in the manuscript, and responses to the reviewers' comments are in the attachment ‘sensors-1850320- coverletter1’. The revised content is marked in red in the revised manuscript.

Reviewer 2 Report

Dear authors,

Your manuscript presents the interesting researchers reefer to monitoring the signal generated by a ball-screw system in order to obtain information on the possible  defects. A complex analysis has been theoretical developed. Also a lot of experiments are realized.

Some comments please accept:

1. In the manuscript and in the experiments are not indicate the type of guide ( sliding or ball recirculating guide).  Also, are not indicate the presence of any lubricant in balls-screw system and in guide system. Presence of the film thickness play an important contribution on the vibration level.

2. The short indications in the table 2,3,4 must be completed, to be understand.

3.In fig. 5 and 7 must be indicate details on the  vertically axis.

Author Response

Dear Reviewer:

Thank you very much for giving us an opportunity to revise our manuscript. We greatly appreciate your constructive comments and suggestions on our manuscript sensors-1850320. The title of the manuscript is ’Ball screw fault diagnosis based on wavelet convolution transfer learning’. Those comments are very helpful for revising and improving our paper, as well as the important guiding significance to other research. We have studied the comments carefully and made corrections which we hope meet with approval. Major corrections are made in the manuscript, and responses to the reviewers' comments are in the attachment ‘sensors-1850320- coverletter2’. The revised content is marked in red in the revised manuscript.

Round 2

Reviewer 1 Report

There are no other questions, other than the suggestion to delete the content of the CWRU data analysis. The authors need to consider that the title of this paper is Ball screw fault diagnosis based on wavelet convolution trans-fer learning, which does not include bearing fault diagnosis.

Author Response

Dear Reviewer:

We greatly appreciate your comments and suggestions on our manuscript sensors-1850320. The title of the manuscript is ’ Ball screw fault diagnosis based on wavelet convolution transfer learning’. Regarding your suggestion to delete the content of the CWRU data analysis, we have carefully studied the comments and made corrections, and the content of the CWRU data analysis has been deleted. The main corrections are in the manuscript and hope to be approved.